# Digitally Fabricated Dentures for Full Mouth Rehabilitation with Zirconia, Polyetheretherketone and Selective Laser Melted Ti-6Al-4V Material

**DOI:** 10.3390/ijerph19053021

**Published:** 2022-03-04

**Authors:** Wei-Ting Lee, Yung-Chung Chen

**Affiliations:** 1Division of Prosthodontics, Department of Stomatology, National Cheng Kung University Hospital, Tainan 704, Taiwan; n105353@mail.hosp.ncku.edu.tw; 2School of Dentistry & Institute of oral medicine, Medical College, National Chen Kung University, Tainan 704, Taiwan

**Keywords:** full mouth rehabilitation, subtractive manufacturing, selective laser melting, removable partial denture, telescopic denture, polyetheretherketone

## Abstract

CAD/CAM technologies have been embedded into the fabrication of removable partial denture (RPD). Various materials such as zirconia and polyetheretherketone (PEEK) are developed for subtractive manufacturing. As for additive manufacturing, dental professionals have begun to use selective laser melting (SLM) techniques for fabricating metallic RPD frameworks. This report demonstrates a case rehabilitated with a maxillary telescopic crown-retained combining PEEK and zirconia material denture and a mandibular Kennedy Class I RPD fabricated with SLM techniques. First, a conventional impression was performed and the master cast was mounted with a centric relation record. Digital models were obtained using tabletop scanners and then the telescopic primary zirconia crowns were designed and milled. After transferring the intraoral distribution of primary crowns using pick-up impression, secondary PEEK crowns and framework were designed, milled, and veneered with composite resin. Mandibular framework was designed and constructed using SLM technique with Ti-6Al-4V. Definitive prostheses for both jaws were finished and delivered. Delivered prostheses functioned well for a one-year period. The was patient satisfied with the improvements in chewing function and esthetics. Both substrative and additive manufacturing techniques are suitable for framework fabrication. Further investigation is needed for improving the mechanical performance and long-term prognosis of digitally made prostheses.

## 1. Introduction

Nowadays, the number of elderly people is rapidly growing and most of them suffer from multiple teeth loss [1]. Removable partial denture (RPD) is a suitable treatment option for them to restore masticatory function, phonetics, and satisfying esthetics [2,3]. Conventionally, frameworks of RPDs are fabricated using lost-wax casting technique with various metallic materials such as cobalt–chromium (Co-Cr) alloy [2]. However, this technique consists of several laboratory procedures, and operational errors which may result in ill-fitted dentures often occur [3].

In addition, telescopic crowns are found to be less traumatic than other retainers, because occlusal forces can be transmitted through the long axis of the abutments [4]. It consists of a primary crown, which is cemented on each prepared tooth to prevent caries and thermal irritation, and a secondary crown, which is rigidly attached to the denture [5,6]. The retention mechanism comes from wedging effect where compressive stress is generated between primary and secondary crowns from occlusal contact. Without using of traditional RPD’s clasps, it provides better esthetics and long-term survival; in particular, the abutments last longer than conventional RPD [6,7].

Recently, computer-aided design and manufacturing (CAD-CAM) workflow has been applied to the laboratory procedures of denture fabrication [3]. Digital models can be obtained using intraoral or tabletop scanners and saved as standard tessellation language (STL) files for further design of RPD frameworks. Final design can be outputted by subtractive manufacturing or additive manufacturing [8]. Dental subtractive manufacturing represents that a fixed-size material block made of ceramic, zirconia, Polyetheretherketone (PEEK), resin, or metal is milled to desired shape. It can provide satisfactory surface precision and dimensional accuracy [9]. PEEK is a member of the Polyaryletherketone (PAEK) family of thermoplastic polymers, consisting of an aromatic backbone molecular chain, interconnected by ketone and ether functional groups [10]. Its relatively stiff backbone allows excellent high-temperature stability, with a melting temperature of 343 °C [6]. Through adequate milling and grinding, PEEK, which provides excellent chemical, mechanical, and thermal properties, has been used for dental implant, implant abutments, dentures, and clamp material [10,11].

On the other hand, additive manufacturing, also known as 3D printing, assembles materials or components in a 3D perspective to form objects from digital models [4]. It can fabricate scrupulous shapes that include undercuts, gaps, and complicated geometries that cannot be easily processed by subtractive manufacturing [3]. The major material combinations for dental purposes include resins, ceramics, and metals [4]. To fabricate metallic prostheses such as RPD frameworks, a selective laser melting (SLM) technique has been used. High-energy laser beam fuses metal powder and 3D objects are constructed with the layer-by-layer manner. Titanium-6 aluminum-4 vanadium (Ti-6Al-4V) was selected for fabricating the presented case RPD framework due to its high strength, hardness, erosion resistance, and biocompatibility [3].

Studies on the performance of PEEK combined with zirconia as a telescopic crown material are still limited [12,13,14,15]. This clinical report describes the clinical application of a maxillary telescopic denture with a PEEK framework supported by primary zirconia crowns and a mandibular RPD framework fabricated with Ti-6Al-4V using SLM technique with one year follow-up.

## 2. Case Description

A 72-year-old man suffered from wobbling anterior maxillary prostheses and severely worn anterior teeth with missing bilateral posterior teeth in the mandible (Figure 1 and Figure 2). After removing maxillary ill-fitted fixed dental prosthesis (FDP) and extracting tooth 22 and 24 due to fractured roots, loss of vertical dimension can be seen from the intraoral examination (Figure 1, Figure 2 and Figure 3, see Appendix A for the detailed settings of camera). The patient stated that he did not want either the palate to be covered by a denture nor implant placement. Therefore, treatment plan was confirmed as maxillary telescopic crowns retained RPD and lower conventional RPD with an ideal centric relation setup and the raised vertical dimension for the patient. Before fabricating definitive prostheses, conventional endodontic treatment for pulp necrosis of teeth 23, 31, 32, 34, 43, and 44 as well as nonsurgical periodontal treatment was completed. Good oral hygiene maintenance was also ensured on the residual teeth and edentulous ridges. Teeth 12, 13, 14, 23, and 25 were prepared with heavy chamfer margin and teeth 31, 32, 33, 34, 42, 43, and 44 were prepared with a heavy chamfer margin on the buccal side and a light chamfer margin on lingual side for installing temporary crowns. The maxillary arch was first classified as Kennedy class IV modification I and then changed to Kennedy class IV after temporary crowns, fixed dental prostheses, and interim dentures were delivered. After wearing the interim prostheses for six months, the patient was satisfied with the esthetic, and the occlusion and temporomandibular joint were stable.

To fabricate definitive prostheses, vinyl siloxanether impressions (Identium, Kettenbach GmbH, Eschenburg, Germany) with stock trays were performed for the maxillary and mandibular abutments. Master casts were poured with the Type IV dental stone (Die-Keen, Heraeus Kulzer, IN, USA) and mounted using facebow transfer and a centric relation interocclusal record made with the bimanual manipulation. The casts and maxillomandibular relationship were scanned using a tabletop scanner (E4, 3shape, Copenhagen, Denmark), and the data were imported into the CAD software (Exocad DentalCAD, exocad GmbH, Darmstadt, Germany) for the designing process. The path of denture insertion was first determined and the telescopic primary crowns were then designed with the taper angle of 6 degrees and thickness of 1 mm. The design of each crown was milled from a zirconia block (VITA YZ-HT, Vita Zahnfabrik, Bad Säckingen, Germany). After the fit of each primary crown was ensured, all of them were picked up using an impression tray (COE metal impression tray, GC America Inc., Alsip, IL, USA) and a vinyl siloxanether material (Identium; Kettenbach GmbH, Eschenburg, Germany) to precisely transfer the relationships of the primary crowns. The impression mold was poured with a Type IV dental stone to form working cast which was digitally scanned for designing the secondary crowns and framework. The design was milled from a PEEK block (JUVORA Dental disc, JUVORA, Lancashire, UK). Milled secondary crowns and framework was treated using sand-blasting with Al2O3 particles of 125-um (White fused alumina, NICHE Fused Alumina, Savoie, French) at a pressure of 2.5 bar and a distance of 10 mm, cleaned in an ultrasonic bath with ethanol for 5 min then air-dried, coated with an adhesive consisting methyl methacrylate, pentaerythritol triacrylate, and photoinitiators (Visio.link, Bredent GmbH & Co KG, Senden, Germay), and finally veneered with the composite resin (Crea.lign, Bredent GmbH & Co. KG, Senden, Germany) for better esthetic (Figure 4 and Figure 5).

The mandibular surveyed crowns were fabricated by lost-wax casting technique. They were cemented permanently to each abutment after successful try-in. The impression with vinyl polysiloxane (Aquasil Ultra; Dentsply Caulk, DE, USA) was made using a customized tray for the mandibular arch. After digitizing the working cast, the RPD framework was designed and constructed using SLM technique with Ti-6Al-4V material (Figure 6). The framework was finished and polished with a series of rubber points and brushes (Metals polishing kit, Shofu Dental Corp, CA, USA), then evaluated and adjusted on the master cast and in the patient’s mouth. The denture teeth were arranged and tested in the patient’s mouth to verify proper tooth position for esthetics, phonetics, and occlusal function. The denture bases were cold-cured processed with acrylic resin (SR Triplex cold, Ivoclar Vicadent AG, Schaan, Liechtenstein).

## 3. Results

After verifying the function, esthetic, and phonetics, definitive prostheses were delivered to the patient. Patient was pleased that the denture was in harmony with the remaining teeth, and his flange was not displayed when he smiled. Home care instructions, including brushing and flossing after each meal, gently brushing the dentures daily, soaking in water overnight, and using denture cleanser three times a week, were given. The patient was asked to return for an appointment the following day and also one week after. The pressure-indicating paste was used to examine the excessive pressure over denture-bearing area and occlusal adjustment of the dentures was performed during the follow-up visit. At the 1-month follow-up, the patient had adapted well to the denture and scheduled routine follow-ups every 3 months. The patient was satisfied with the retention and chewing function of the dentures. He was able to eat nuts and steak, which is equivalent to score 4 of the chewing ability index [16]. During the 1-year observation period, no specific mobility of the abutment teeth, and the prostheses were functioning properly. The patient was satisfied with the treatment, and the entire prognosis of the rehabilitation is expected to be good (Figure 7, Figure 8 and Figure 9).

## 4. Discussion

Multiple teeth missing and unfavorable abutment distribution over the maxillary region makes this case challenging to be restored. Considering that the patient was not willing to receive implant placement and demanded a palatal-free prosthesis, the alternative treatment plan was a long-span FDP. However, excessive deformation under occlusal loads may cause failure of a long-span FDP. It can lead to fracture of a porcelain veneer, breakage of a connector, or loosening of a retainer [17,18]. The survival estimate for long-span FDPs is 88.3% after 5 years and 73.2% after 10 years [19]. Telescopic system can help with preserving the abutment teeth, and stabilize existing mobile teeth. This can be attained by secondary splinting of rigid connection between the abutment teeth and the telescopic crowns [20]. The mean survival rates for abutment teeth of telescopic dentures ranged from 60.6% to 95.3% after an observation period of 4 to 10 years [21]. Abutment teeth loss was attributed mainly to progression of periodontal disease, secondary caries, and tooth fractures. The survival rates of the telescopic dentures were found to be between 66.7% and 98.6% after an observation period of 6 to 10 years [22]. The major complications include loss of abutment ranging from 20.6% to 37%, facing repair ranging from 11.1% to 26.9% and denture reline from 12% to 20.6%. However, regular follow-up enables minor damage of delivered dentures to be noticed and adjusted to increase the long-term survival [20,23].

The use of PEEK and lirconnia material can be excellently embedded in the CAD/CAM workflow. Relatively low flexural modulus of PEEK material (4 Gpa) resulted in a growing retention between primary and secondary crowns in 10 and 20 groups comparing to the 0o group [13]. Considering its unfavorable periodontal support in the present study, a larger taper angle was used for the primary telescopic crown on the maxillary lateral incisor to reduce its retention force. If the clinical try-in step of secondary crowns and the framework is not successful, the framework needs to re-fabricated and hence the laboratory fee increases remarkably. In contrast to the conventional way, the authors used self-curing acrylic resin to fabricate a trial denture which allows detailed adjustment and confirmed the adequate geometry of the secondary crowns and the framework. With the help of the CAD/CAM workflow, the STL files can be saved and efficiently refabricate primary crowns or framework if the denture needs to be repaired in the future. Comparing with gold alloy (91.5 GPa) and Co–Cr alloy (207 Gpa), elastic modulus of PEEK material was relatively low. Therefore, to obtain adequate rigidity for a framework, the thickness of PEEK material needs to be increased [24]. In addition, PEEK have a grayish-brown or pearl-white opaque color and need to be veneered with a composite resin, especially for the anterior region, but it might barely satisfy patients’ esthetic demands due to the thicker PEEK material combining with the resin. Most studies concluded that using adhesive systems containing methylmethac-monomers (MMA) such as Signum PEEK bond and Visio.link is more clinically reliable [25,26,27,28].

Utilizing the SLM technique for fabricating RPD framework has raised lots of clinicians’ attention. Bajunaid and his colleagues reported that SLM-fabricated RPD frameworks exhibited better fit accuracy than conventional lost-wax casting technique [2]. Peng and her colleagues compared the trueness of RPD framework by casting with Co-Cr, SLM with Ti-6Al-4V, and SLM with Co-Cr, and indicated that the smallest surface deviations was observed in the group of SLM with Ti-6Al-4V [3]. Nevertheless, due to lower elastic modulus (110GPa) of Ti-6Al-4V which is about one half of the cobalt-chromium alloys, careful consideration should be given to design Ti-6Al-4V RPD framework, especially for clasp assemblies [29]. The retentive clasp made by cast Co-Cr alloys, the terminal third of an occlusally approaching retentive clasp should go across the survey line and engage undercut of 0.25 mm deep to the gingival third with an uniform tapered from its point of attachment to the tip [30]. Previous studies have suggested that 5~9 N is required for the clinical use of an RPD clasp [31,32,33]. The fatigue test was performed for evaluating the RPD clasp retentive force fabricated by SLM and casting with Ti-6Al-4V under conditions of engaging undercuts of 0.25 mm and 0.5 mm. The clasps were moved perpendicularly on and off the abutment teeth for 7200 cycles, the result showed that the final retentive force of the SLM group engaging undercut of 0.25 mm is higher than 5 N, and engaging undercut of 0.5 mm is higher than 6 N. SLM groups had better fatigue resistance than casting group and it could be engaged in the undercut of 0.5 mm [34]. Furthermore, a comparison of remaining retentive forces in titanium clasp fabricated by SLM, CNC milling and casting after 10,000 insertion/removal cycles showed that the retentive force of SLM Ti clasp was 11.79N at the beginning and is significantly higher than other groups, but it rapidly declined until fracture at 4000 cycles [35]. The reliable information of clasps SLM with Ti-6Al-4V on their clinical application is still not available. The design for clasps fabricated with Ti-6Al-4V by the SLM technique is surely different from that of the conventional casting Co-Cr alloy, and it needs further investigation for the microstructure and properties of RPD clasps. The major limitation of the present report is that the follow-up period was not long enough to further evaluate the long-term prognosis of digitally made prostheses.

## 5. Conclusions

CAD/CAM technologies for fabricating PEEK frameworks of a telescopic denture and a RPD framework SLM with Ti-6Al-4V are successfully applied for a case of full mouth rehabilitation. After a year of clinical follow-up, the prostheses have functioned well and all abutment teeth are periodontally stable without any complications. Longer-term follow-up is necessary and a related investigation should be conducted to ensure the clinical application of this prostheses.

## Figures and Tables

**Figure 1 ijerph-19-03021-f001:**
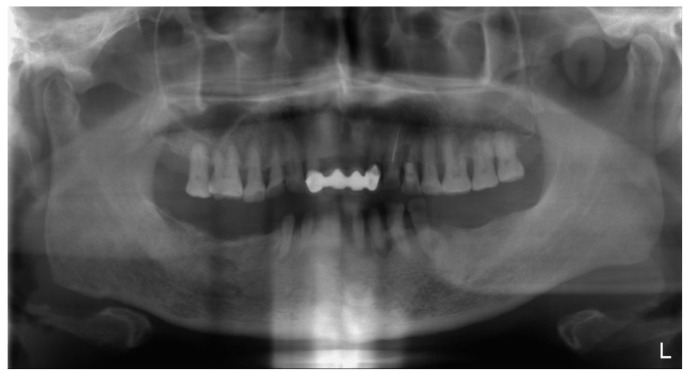
Initial panoramic film.

**Figure 2 ijerph-19-03021-f002:**
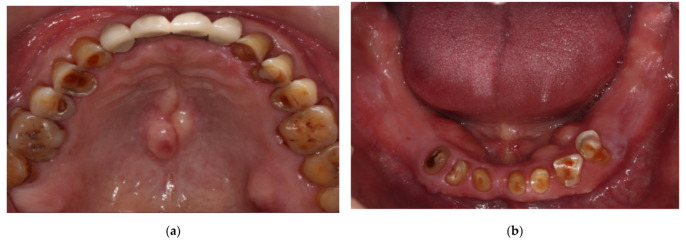
(**a**) Occlusal view of the maxillary arch at the beginning; (**b**) Occlusal view of the mandibular arch at the beginning.

**Figure 3 ijerph-19-03021-f003:**
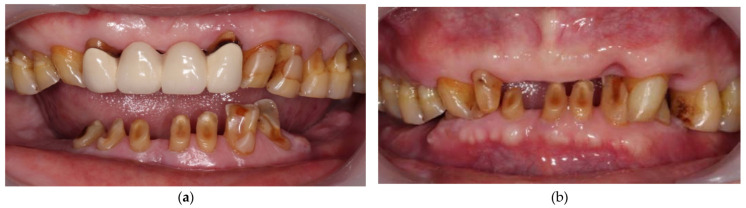
(**a**) Frontal view at the beginning of the treatment; (**b**) Frontal view after removing maxillary ill-fitted prosthesis and extracting hopeless teeth.

**Figure 4 ijerph-19-03021-f004:**
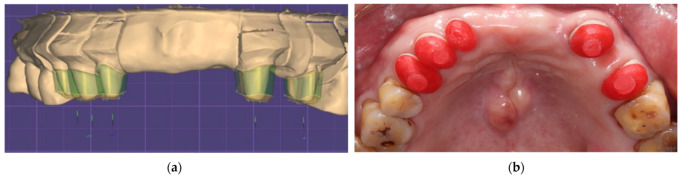
(**a**) Digital design of telescopic primary crowns; (**b**) intra-oral try-in of telescopic primary crowns and resin pattern caps were made for pick-up impression.

**Figure 5 ijerph-19-03021-f005:**
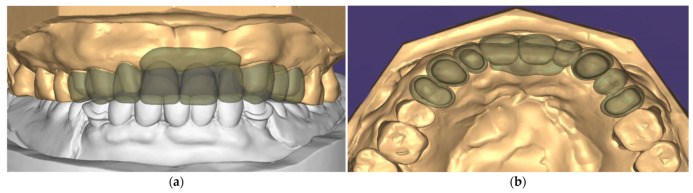
(**a**) Frontal view of the digital design of telescopic secondary crowns and framework; (**b**) Occlusal view of the digital design of telescopic secondary crowns and framework.

**Figure 6 ijerph-19-03021-f006:**
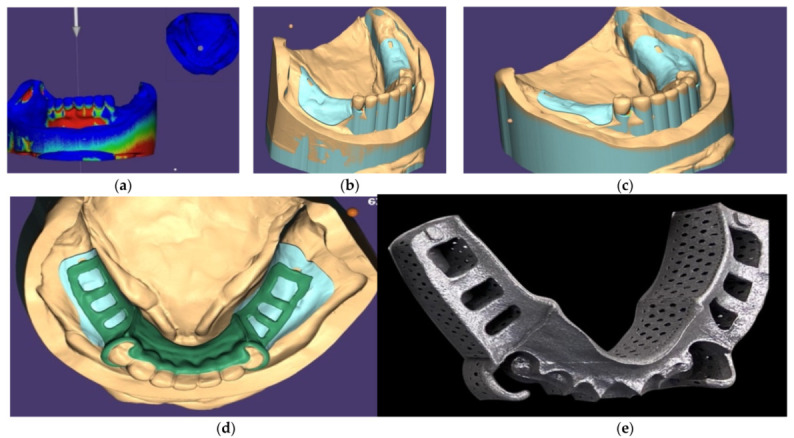
(**a**) Digital survey for determining path of insertion of the lower denture framework; (**b**,**c**) Blocking out unwanted undercut; (**d**) Designing framework using CAD software; and (**e**) Constructing framework using SLM technique with Ti-6Al-4V material.

**Figure 7 ijerph-19-03021-f007:**
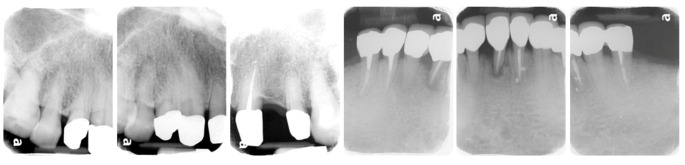
Periapical films of the definitive prostheses.

**Figure 8 ijerph-19-03021-f008:**
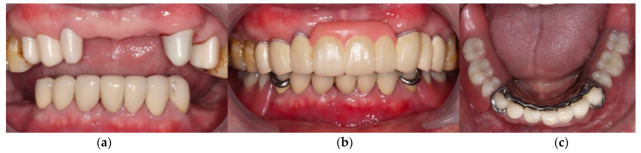
The definitive prostheses delivered to the patient. (**a**) Frontal view of primary telescopic crowns; (**b**) Frontal views of the whole prosthesis; (**c**) Occlusal view of the mandibular RPD.

**Figure 9 ijerph-19-03021-f009:**
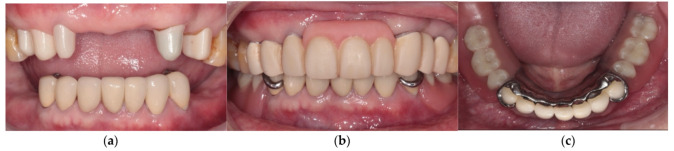
The prostheses delivered to the patient after 1 year. (**a**) Frontal view of primary telescopic crowns; (**b**) Frontal views of the whole prosthesis; and (**c**) Occlusal view of the mandibular RPD.

## Data Availability

Not applicable.

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
