# Peer review of "Digitally Fabricated Dentures for Full Mouth Rehabilitation with Zirconia, Polyetheretherketone and Selective Laser Melted Ti-6Al-4V Material"

_ijerph, 2022, doi:10.3390/ijerph19053021_

Round 1
Reviewer 1 Report
Summary of manuscript: This manuscript reports the fabrication of the dentures with zirconia, polyeheretherketone and selective laser melted Ti-6Al-4Vmaterial. This report concludes that CAD/CAM technologies for applying PEEK frameworks of a telescopic denture and a RPD framework SLM with Ti-6Al-4V are suitable for framework fabrication.
General comments
- There is no functional evaluation and aesthetic evaluation of the fabricated dentures.
- There is no novelty about PEEK combined with zirconia as a telescopic crown material because some studies have been reported.
- There are no photos of one year after wearing new dentures and is no current panoramic X-ray photograph.
Author Response
The response letter has been uploaded.

Reviewer 2 Report
- Please eliminate the subtitles in the abstract - Line 36: Co-Cr are never defined in the text, please do - Line 42: rigidly? you have to use another term i think- Line 43-46 need a reference - Line 72: please add a space between "limited" and "[10-13]". - Lines 71-75: please add the clinical period to the aim of the study - Line 85: "periodontal treatment was completed", How?
- Please clarify the originality of this study in the introduction part
- Figure 1: not clear, more resolution is recommended
- Figure 2: please add (a) and (b) for the figure parts and explain the figure legend - What was the camera which was used to take the intraoral photos? please add in the material methods part - Line 108: please eliminate the space 1mm. The design
- Lines 109-110: please rephrase, not clear
-Line 117: drying after ethanol ? of waiting some min? - Line 117: Which kind of adhesive? universal? generation 7, please more details
- Line 134: please more info on oral hygiene instructions - Figure 5: same comment for figure 2 - Discussion well written, Line 221: please eliminate the space between [32]. and The reliable - Please add the limitation of this study at the end of discussion part
- Please follow MDPI style for all the references
- Please add Institutional Review Board Statement
Author Response
The response letter has been uploaded.

Reviewer 3 Report
The authors describe the use of modern techniques and materials for the full rehabilitation of the oral cavity. It is especially important for practicing dentists because it shows new possibilities and technological solutions. The article is also interesting because of the growing number of elderly patients.
The prosthetic part of the case report is very carefully prepared. Each stage of the work is described in detail and well illustrated. However, I would like to ask you to complete the part concerning the stage of patient oral cavity preparation. The description should contain information about which teeth were qualified for extraction and for what reasons. Were the teeth intended under crowns endodontically treated? The authors only mention the end of periodontal treatment, what kind of treatment? There is also no information about patient recommendations for home hygiene procedures.
I believe that the article is suitable for publication after taking into account the above small corrections
Author Response
The response letter has been uploaded.

Round 2
Reviewer 2 Report
The authors made a good job